# Genomic divergence during feralization reveals both conserved and distinct mechanisms of parallel weediness evolution

Toshiyuki Imaizumi [1✉], Kaworu Ebana[2], Yoshihiro Kawahara[3], Chiaki Muto[2], Hiroyuki Kobayashi[4,6], Akira Koarai[1] & Kenneth M. Olsen [5]

Agricultural weeds are the most important biotic constraints to global crop production, and chief among these is weedy rice. Despite increasing yield losses from weedy rice in recent years worldwide, the genetic basis of weediness evolution remains unclear. Using whole-genome sequence analyses, we examined the origins and adaptation of Japanese weedy rice. We find evidence for a weed origin from *tropical japonica* crop ancestry, which has not previously been documented in surveys of weedy rice worldwide. We further show that adaptation occurs largely through different genetic mechanisms between independently-evolved *temperate japonica*- and *tropical japonica*-derived strains; most genomic signatures of positive selection are unique within weed types. In addition, some weedy rice strains have evolved through hybridization between weedy and cultivated rice with adaptive introgression from the crop. Surprisingly, introgression from cultivated rice confers not only crop-like adaptive traits (such as shorter plant height, facilitating crop mimicry) but also weedy-like traits (such as seed dormancy). These findings reveal how hybridization with cultivated rice can promote persistence and proliferation of weedy rice.

[1] Institute for Plant Protection, National Agriculture and Food Research Organization, Tsukuba, Japan. [2] Research Center of Genetic Resources, National Agriculture and Food Research Organization, Tsukuba, Japan. [3] Research Center for Advanced Analysis, National Agriculture and Food Research Organization, Tsukuba, Japan. [4] Central Region Agricultural Research Center, National Agriculture and Food Research Organization, Tsukuba, Japan. [5] Department of Biology, Washington University in St. Louis, St. Louis, USA. [6] Center for Weed and Wildlife Management, Utsunomiya University, Utsunomiya, Japan. ✉email: toima@affrc.go.jp

Agricultural weeds are the most important biotic constraints to global crop production[1]. Estimated crop losses from weeds in the US and in Australia exceed 33 billion US dollars and 3.3 billion Australian dollars, respectively[2,3]. Despite the severe economic impact they impose, the mechanisms underlying the origin and spread of agricultural weeds remain largely unresolved. Agricultural weeds can evolve from both wild and domesticated plant species, and some of the most problematic agricultural weeds are close relatives of domesticated crops[4]. Chief among these is weedy rice (*Oryza* spp.). This relative of cultivated Asian rice has emerged as a dominant weed species in paddy rice fields worldwide, creating serious global agricultural problems in recent decades[5]. In the US alone, yield losses from weedy rice could feed an additional 12 million people per year[6].

Recent evidence from whole-genome resequencing indicates that weedy rice has evolved independently in multiple regions worldwide, and that it has largely evolved by de-domestication (feralization) from three different cultivated rice subgroups: *temperate japonica* (hereafter, TEJ), *indica* (IND) and *aus* (AUS)[7–11]. Evolution from the other two major rice subgroups, *tropical japonica* (TRJ) and *aromatic* (ARO), has not been previously reported. In a recent study[11], genome scans for positive selection indicated that weedy rice adaptations have little overlap with loci targeted during rice domestication, pointing to the involvement of different sets of genes during rice domestication and de-domestication. However, many other aspects of weedy rice evolution remain unclear, including the extent to which their repeated parallel evolution from cultivated rice has occurred through shared or different genetic mechanisms.

Studies of parallel evolution and adaptive introgression between independently evolved groups can provide valuable insights into the genetic mechanisms that underlie adaptation and the potential constraints on this process[12–14]. Weedy rice is an especially good model system for studying these mechanisms given its very close relationship to the genomic model system rice, and its evolutionary history of multiple independent origins from different crop varieties[15]. In addition, while both cultivated and weedy rice are primarily self-pollinating, outcrossing occasionally occurs, and the proximity of weed strains and crop varieties within rice fields can facilitate hybridization and adaptive introgression into weed populations, including for traits such as herbicide resistance[16–18].

Diverse weedy rice populations have been documented in all of the world's major rice-growing regions[7–11]. In Japan, where weedy rice has emerged as a major problem in the last two decades[19], there are two major weed morphotypes: black hull (BH) and straw hull (SH) (Fig. 1a). Both are closely related to *japonica* rice varieties (the major cultivars in Japan) but evolved through independent feralization events[11]. Both strains possess typical weedy rice traits, including a red pericarp, easily shattered seeds and persistent seed dormancy[20]; these traits are all characteristic of wild *Oryza* species and were selected against during rice domestication. BH weedy rice also has several additional wild-like traits, including taller plant height than the crop, a red-pigmented apiculus and a black hull (Fig. 1b, c, Supplementary Data 1). SH weedy rice has more crop-like traits; in addition to a straw hull, these include shorter plant height and, in some cases, a non-pigmented apiculus[21]. This combination of weedy traits and traits shared with modern cultivars suggests that the SH weed morphotype could have evolved through hybridization between weedy and cultivated rice. The independent evolution of the BH and SH weed strains, both from *japonica* rice ancestors, provides a unique opportunity to study the genetic basis of weediness evolution and the extent to which it occurs through conserved or distinct mechanisms. Moreover, to the extent that SH weeds are

of crop-weed hybrid origin, these strains can also provide a valuable system to study weed adaptation via cultivated rice introgression.

Here, we analysed whole-genome sequences of weedy and cultivated rice in Japan to characterize the genomic basis of evolution in the BH and SH weed strains. Our analyses reveal that these weeds have largely evolved through de-domestication from cultivated *temperate japonica* (TEJ) rice and that, unlike other world regions, some strains have also evolved from *tropical japonica* (TRJ) varieties. We also detected a weed-crop hybrid origin for some SH strains, whose very close relationship to cultivated rice provided a rare opportunity to identify the genomic regions that are most critical for rice feralization. Surprisingly, we find that genetic contributions from cultivated rice confer not only crop-like weed-adaptive traits (such as shorter plant height, facilitating crop mimicry) but also wild-like adaptations (including deep seed dormancy). Our findings reveal how feralization and subsequent hybridization with cultivated rice can promote persistence and proliferation of its weedy descendants.

## Results

**Population structure of weedy rice**. Short-read whole-genome sequencing of 50 contemporary Japanese weedy rice strains, five weedy rice strains collected in the 1970s, and 33 Japanese landraces yielded 22.6× average genome coverage (Supplementary Data 1). These data were analysed together with previously published whole-genome sequences for 86 cultivated and weedy rice strains selected on the basis of high genome coverage; 78% of the selected strains had >15× genome coverage (the minimum genome coverage was 11.8×). Genotype likelihood generated from 23.0× average genome coverage sequences and a combined genotype dataset of 13,315,365 raw variants (11,476,514 SNPs and 1,838,851 indels) across 174 samples were used in the population genomic analysis described below.

Principal component analysis (PCA) and phylogenetic analysis indicated that weedy rice in Japan shares ancestry predominantly with varieties of the TEJ and TRJ groups of cultivated rice. PCA separated cultivated rice varieties into three groups: TEJ, TRJ and IND + AUS (Fig. 1d). The first principal component explained 96% of the total variation and distinguished the *japonica* rice subgroups (TEJ and TRJ) from the *indica* rice subgroups (IND and AUS). Only 2% of the variance was accounted for by the second principal component separating TEJ and TRJ.

All of the BH weedy rice strains collected in Japan belonged to the TEJ group (Fig. 1d, lower left corner), and SH weedy rice was grouped with either the TEJ or TRJ group (lower left or upper left corner, respectively); (Fig. 1d, Supplementary Data 1). A phylogenetic tree based on 1,936,292 homozygous SNPs further confirmed that weedy rice strains in Japan belonged to the TEJ or TRJ groups with high bootstrap support (Fig. 1e, Supplementary Fig. 1). On the basis of these results, Japanese weedy rice strains could be classified into three groups: TEJ-derived BH weedy rice, TEJ-derived SH weedy rice, and TRJ-derived SH weedy rice (hereafter, BH, SH_TEJ and SH_TRJ, respectively). SH_TEJ strains were further classified into two subgroups in the phylogenetic tree with high bootstrap support, and the subgroups were consistent with the classification by apiculus colour (Supplementary Data 1): SH1_TEJ (red-pigmented apiculus) and SH2_TEJ (non-pigmented apiculus) (Fig. 1d and e).

To assess population structure among the Japanese weedy rice strains and other samples, we performed admixture analysis based on NGSadmix. Evaluation of the likelihood of $K$ ($L(K)$)[22] and $\Delta K$[23] suggested $K = 5$ as the best model (Supplementary Fig. 2). Genetic subgroups at $K = 5$ corresponded to the following groups of accessions: (1) modern_TEJ and landrace_TEJ plus

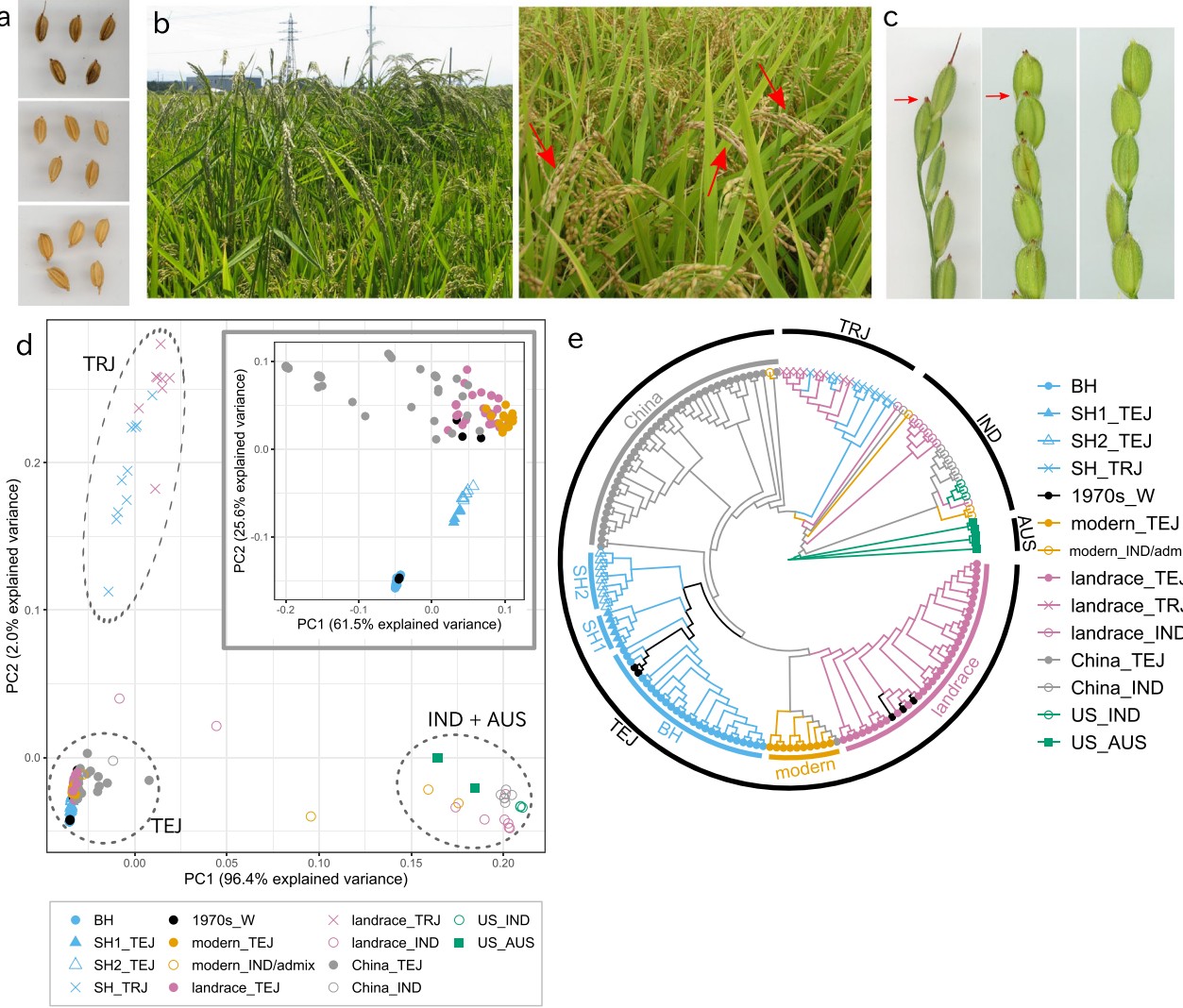

**Fig. 1 Morphotypes and genomic classification of Japanese weedy rice. a** Hull colour of two major morphotypes of Japanese weedy rice and cultivated rice (Koshihikari): BH weedy rice (top); SH weedy rice (centre); and Koshihikari (bottom). **b** BH (left) and SH (right) weedy rice in a paddy rice field (red arrows indicate spikelets of SH weedy rice). **c** Spikelets of BH weedy rice with a red-pigmented apiculus (left), SH weedy rice with a red-pigmented apiculus (centre), and SH weedy rice with a non-pigmented apiculus (right). **d** PCA plot of 174 rice strains, including weedy rice from Japan, China and the US and landrace and modern cultivated rice from Japan; the first and second eigenvectors were obtained using genotype likelihoods estimated by ANGSD using the aligned and mapped reads of whole-genome sequencing data. PCA plot of 129 TEJ rice strains including BH, SH_TEJ, modern_TEJ, landrace_TEJ and China_TEJ, showing the first and second eigenvectors in a subplot (top right corner). **e** Phylogenetic tree based on 1,936,292 homozygous SNPs. TEJ, *temperate japonica*; TRJ, *tropical japonica*; IND, *indica*; AUS, *aus*; BH, TEJ-derived BH weedy rice; SH1_TEJ, TEJ-derived SH weedy rice with a red-pigmented apiculus; SH2_TEJ, TEJ-derived SH weedy rice with a non-pigmented apiculus; SH_TRJ, TRJ-derived SH weedy rice; 1970s_W, weedy rice collected in the 1970s in Japan, China_TEJ and China_IND, Chinese TEJ- or IND-derived weedy rice; US_IND and US_AUS, US IND- or AUS-derived weedy rice.

three of five weed strains collected in the 1970s; (2) landrace_TRJ and the SH_TRJ weed strains; (3) BH weeds, including two strains collected in the 1970s; (4) weedy rice from China; and (5) modern_IND/admix, landrace_IND and US_IND/AUS, along with some weedy rice from China (Fig. 2a; Supplementary Data 2). The fact that two of the five weedy rice strains collected in the 1970s belonged to the same group as present-day BH weedy rice (Fig. 1d, e and Fig. 2a) suggests that the now-pervasive BH strains have existed since at least the 1970s. Whereas the BH and SH_TRJ weedy rice did not exhibit admixed ancestry, the SH_TEJ weedy rice showed admixed ancestry between BH weedy rice and modern/landrace_TEJ cultivated rice from Japan (Fig. 2a; Supplementary Data 2); this is consistent with the SH_TEJ weeds having evolved through hybridization of BH weeds and TEJ crop varieties.

To further assess the relationships of weedy rice with specific cultivars, kinship analysis was carried out using the same samples used in population structure analysis, plus 11 cultivated rice strains (Supplementary Data 1) that were previously shown to have close kinship with Japanese weedy rice[11]. BH weedy rice strains showed close kinship (0.7–0.85) with other BH strains but low kinship (0.3–0.4) with contemporary TEJ cultivated rice (Supplementary Fig. 3); notably, this includes Ssal Byeo, a Korean rice landrace which had been found to have the closest kinship with Japanese TEJ-derived weedy rice in the previous study[11]. This result suggests that unlike many weedy rice strains in other parts of Asia, Japanese BH weedy rice is not a descendant of contemporary rice cultivars but rather is more likely to have evolved from traditional landraces that are no longer cultivated in Japan. In contrast, SH_TRJ weedy rice strains showed close

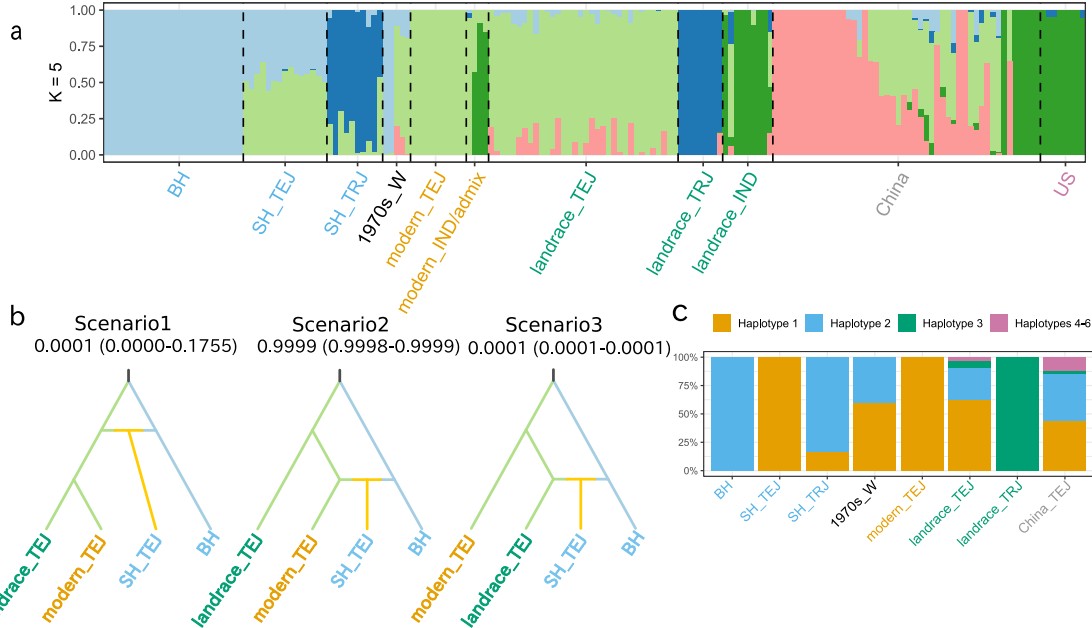

**Fig. 2 Origin of Japanese weedy rice. a** Population structure of weedy and cultivated rice strains. Ancestry proportions for individuals with $K = 5$ are presented. **b** Demographic scenario models involving BH and SH_TEJ weedy rice and modern and landrace cultivated rice for deducing the hybridization history of SH weedy rice. Scenario 1: SH weedy rice evolved from a hybridization event between BH weedy rice and cultivated rice before the divergence of modern and landrace strains; Scenario 2: it evolved from a hybridization event between BH weedy and modern cultivated rice; and Scenario 3: it evolved from a hybridization event between BH weedy and landrace cultivated rice. Posterior Probability (95% of confidence interval) for each scenario was indicated under the title of each scenario. **c** Relative frequency of chloroplast genome haplotypes in weedy and cultivated rice groups. Variants of each of the chloroplast genome haplotypes are indicated in Supplementary Table 1.

kinship (0.6–0.7) with contemporary TRJ cultivars from Japan and Korea. SH_TEJ weedy rice strains had intermediate kinship with BH weedy rice and cultivated rice from Japan (0.45-0.55), consistent with descent from BH-TEJ hybrids. The kinship values also supported the PCA results where the SH weedy rice strains were plotted between the BH weedy and cultivated rice strains from Japan (Fig. 1d).

**Demographic modelling confirms a crop-weed hybrid origin for SH_TEJ weeds.** Statistical assessment of four potential demographic scenarios for the origin of the putative hybrid-derived SH_TEJ weedy rice (Supplementary Fig. 4a) revealed overwhelming support for one scenario: emergence of SH_TEJ weedy rice from a hybridization event between BH weedy rice and cultivated rice (posterior probability: 0.8721; 95% confidence interval (CI) = 0.8633–0.8810). Having established this hybrid origin for the weedy strains, we next compared demographic scenarios to assess the timing of the hybridization event with three proposed scenarios (Fig. 2b). Model analysis revealed acceptable model fits for scenarios 1 and 2 (Supplementary Fig. 5), with scenario 2 best supported according to the posterior probability (0.9999 with 95% CI = 0.9998–0.9999). Under this scenario, SH_TEJ weedy rice is inferred to have originated through hybridization between BH weedy rice and modern TEJ rice or their parental pedigree landrace varieties.

To identify the maternal origin of the hybrid-origin SH_TEJ weedy rice, variants in chloroplast genome sequences were compared. Chloroplast genomes of 130 TEJ and TRJ cultivated and weedy rice strains could be classified into six haplotypes (Supplementary Table 1), with most accessions carrying haplotype 1 or 2 (Fig. 2c and Supplementary Data 1). All BH strains carried haplotype 2, and all SH_TEJ strains carried haplotype 1. The sharing of haplotypes 1 by all SH_TEJ weeds and all modern TEJ cultivars suggests a maternal TEJ crop origin for these weeds.

Considering the admixture and demographic results together (Fig. 2a and b), the origin of the SH_TEJ weeds thus appears to have been by pollen from BH weedy rice plants fertilizing cultivated TEJ plants, which then produced seeds that gave rise to the hybrid weedy genotype. For TRJ crops and weeds, landrace_TRJ strains carried haplotype 3, but most of SH_TRJ weeds carried haplotype 2, which is shared with BH and landrace_TEJ. As this could suggest a possible hybrid origin for the SH_TRJ weeds, we conducted statistical assessment of potential demographic scenarios for their origin; however, demographic modelling did not support the possibility of hybrid origin between landrace_TRJ and BH or landrace_TEJ (Supplementary Fig. 4b).

**Genomic regions under selection in weedy rice.** Adaptations that arose during weedy rice evolution and that distinguish it from its crop ancestor should be associated with genomic regions that show high genetic differentiation in weed-crop comparisons. To identify such adaptively deriverged regions, genomic scans of differentiation were performed using the population differentiation index $F_{ST}$. Focusing first on the weed strains that were not of admixed ancestry, BH was compared to TEJ, and SH_TRJ was compared to TRJ (Fig. 3a, c). BH/TEJ showed significant differentiation in 162 regions (19.5M bp in total, representing 5.21% of the genome), and SH_TRJ/TRJ showed significant differentiation in 96 regions (27.8M bp in total, representing 7.44% of the genome). Comparing the two outlier scans, 16 outlier regions (3.3M bp, 0.88% of the genome, Supplementary Fig. 6) were shared between scans. This low proportion of shared outliers suggests that weediness adaptation largely occurred through different genetic mechanisms between TEJ- and TRJ-derived weedy rice. This finding parallels a previous finding of different underlying genetic mechanisms for de-domestication in IND- and AUS-derived weedy rice in the US[7,24]. In another study of Asian

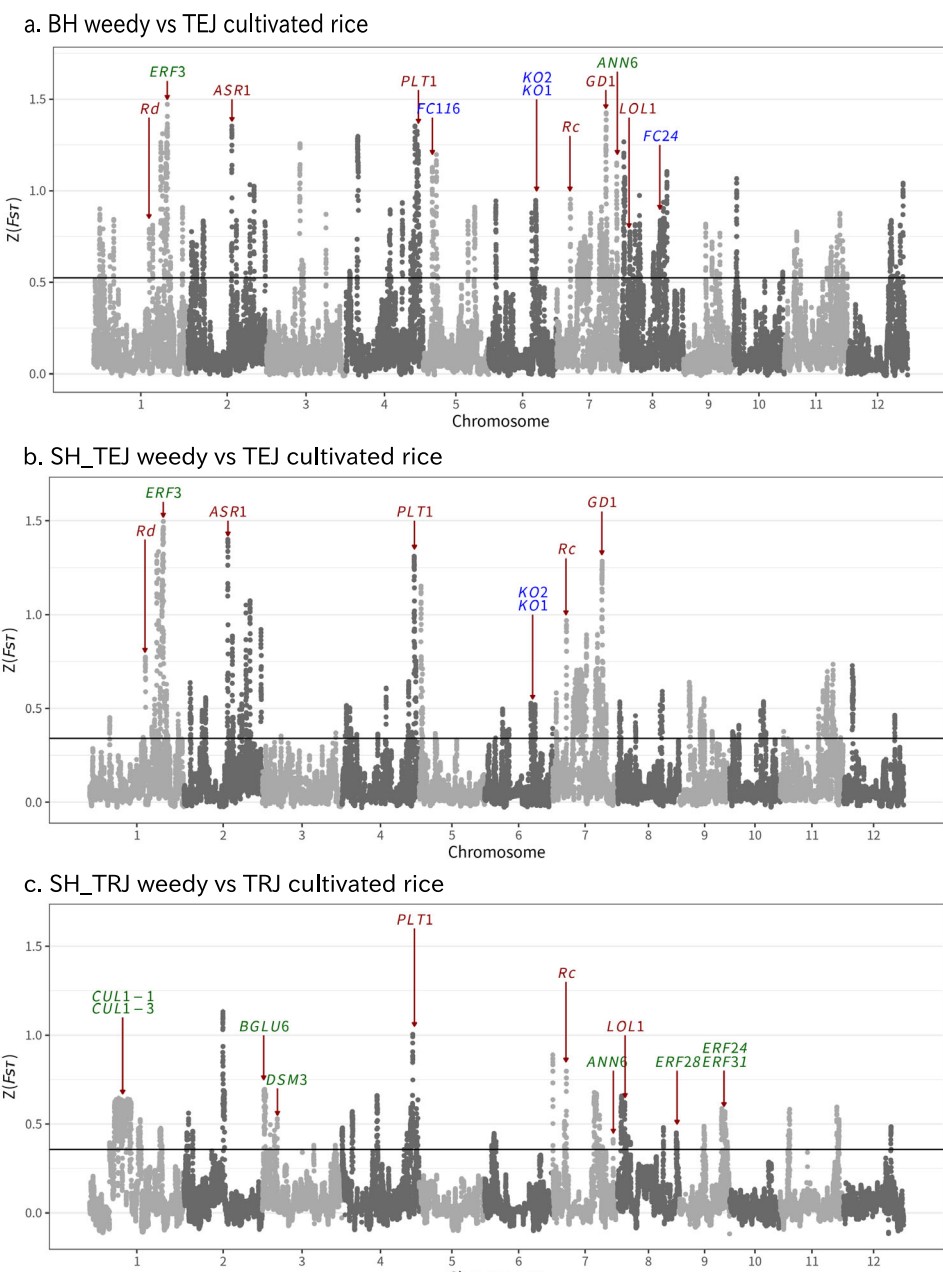

**Fig. 3 Genomic regions under selection in weedy rice.** $F_{ST}$ ($Z$-transformed) between TEJ cultivated rice and BH and SH_TEJ weedy rice (**a**, **b**, respectively) and between TRJ cultivated and SH_TRJ weedy rice (**c**). TEJ cultivated rice includes both modern and landrace strains. Each dot is a 100-kb window with a step size of 10 kb, and the solid line indicates the cut-off for the threshold of five absolute median deviations. Significant windows that were located continuously were integrated to outlier regions, and 162, 133 and 96 outlier regions were detected in (**a**), (**b**) and (**c**), respectively. Some functionally known genes locatied in the significant regions are shown: that regulating a red pericarp colour and/or seed dormancy in dark red, that regulating drought tolerance in green, and that regulating plant height and/or endogenous bioactive gibberellins in blue.

weedy rice[11], the seed germination-related gene *GD1* on chromosome 7, was included in the significantly differentiated region between TEJ-derived weedy and cultivated rice; however, in our study, the gene was not included in the differentiated region between SH_TRJ weedy and TRJ cultivated rice, and some SH_TRJ weedy rice had a cultivated-type allele (Fig. 4a; Supplementary Table 2; Supplementary Data 3). This finding further supports that weediness adaptation occurs through different genetic mechanisms between TEJ- and TRJ-derived weedy rice.

For BH weedy rice, multiple genes associated with plant height were detected in significantly differentiated regions compared with its cultivated ancestor (Fig. 3a). The regions contain

*FRAGILE CULM 24* (*FC24*) and *116* (*FC116*), the mutants of which show reduced plant height[25,26]. BH weedy rice strains have some polymorphic sites, including nonsynonymous SNPs in *FC116* compared with those in Nipponbare (Fig 4a; Supplementary Table 2; Supplementary Data 3). A cluster of *ent-Kauren oxidase* (*KO*) genes associated with gibberellic acid (GA) synthesis and semidwarf plant height[27–29] were detected in significantly differentiated regions between BH weedy and cultivated rice. For SH_TRJ weedy rice, several genes regulating drought tolerance were detected in significantly differentiated regions compared with TRJ cultivated rice (Fig. 3c). Shared differentiated regions in BH and SH_TRJ weedy rice include *Rc* (a

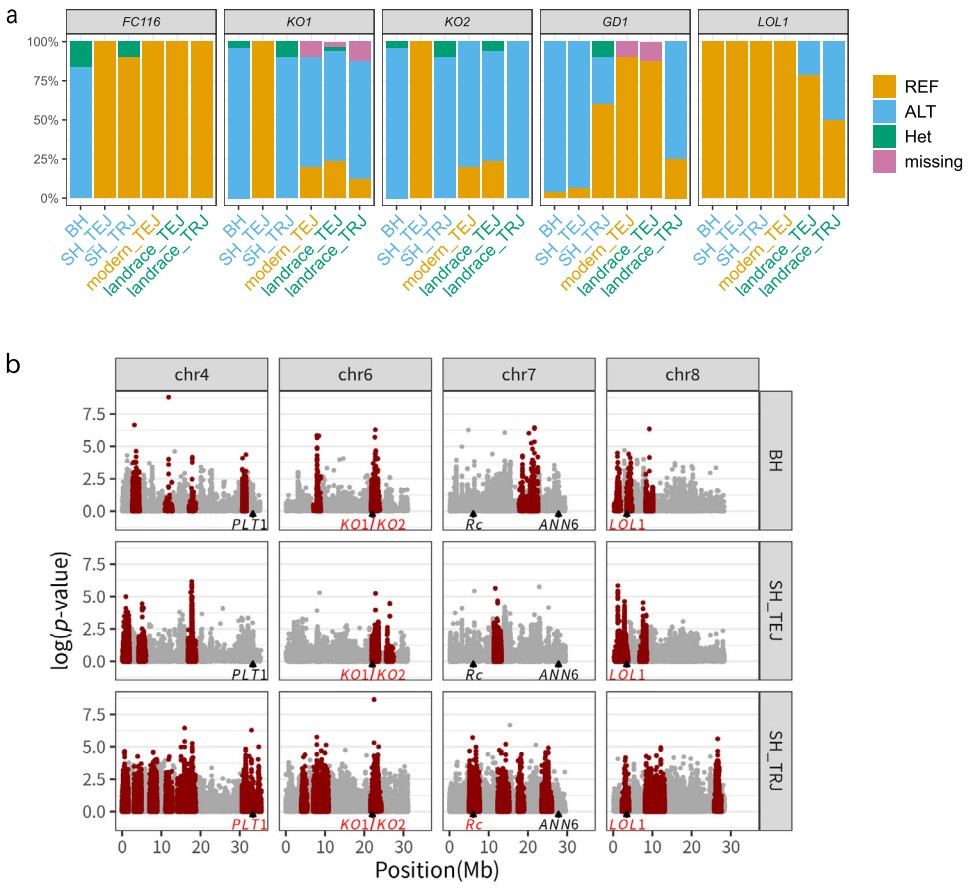

**Fig. 4 Characterization of the significantly differentiated genes regulating plant height and/or endogenous bioactive gibberellins. a** Frequency of variants for nonsynonymous variants in *FC116*, *KO1*, *KO2*, *GD1* and *LOL1*. Detailed information is provided in Supplementary Table 2. BH, frequency in 25 BH weedy rice strains; SH_TEJ, frequency in 15 SH_TEJ weedy rice strains; SH_TRJ, frequency in 10 SH_TRJ weedy rice strains; modern_TEJ, frequency in 10 modern_TEJ cultivated rice strains; landrace_TEJ, frequency in 33 landrace_TEJ cultivated rice strains; landrace_TRJ, frequency in 8 landrace_TRJ cultivated rice strains. **b** Plot of iHS score (the associated *p*-values are indicated instead of the iHS score itself) against physical distance for chromosomes 4, 6, 7 and 8 to identify regions displaying strong signatures of positive selection in the BH (top), SH_TEJ (centre) and SH_TRJ (bottom) weedy rice groups. Regions under positive selection according to a threshold of $p < 0.0001 (-\log10(p\text{-value}) > 4.0)$ are shown in dark red. The black triangles on the bottom correspond to candidate genes (*PLT1*, *KO1*, *KO2*, *Rc*, *ANN6* and *LOL1*) located in shared differentiated regions between BH and SH_TRJ or BH and SH_TRJ as identified by $F_{ST}$ values (Fig. 3), and candidate genes in the regions under positive selection as detected by the iHS test are shown in dark red.

regulatory gene that pleiotropically controls both the pericarp and seed dormancy[30]) on chromosome 7[11], *LOL1* (regulating GA synthesis) on chromosome 8, *PLT1* (regulating root development[31] and stress response[32]) on chromosome 4, and *ANN6* (regulating drought tolerance) on chromosome 7 (Fig. 3a, c). Together these patterns suggest that weed adaptation may be occurring through mechanisms involving seed dormancy and stress tolerance.

As an apparent hybrid derivative of TEJ cultivars and BH weeds, which themselves descended from TEJ cultivars, the highly crop-like SH_TEJ weed strain provides a unique opportunity to identify the genomic regions that are most critical for feralization. Specifically, the subset of loci that are differentiated between BH and TEJ and that remain differentiated between SH_TEJ and TEJ should be enriched for loci that are required for the adaptation of a weedy life history. Comparing SH_TEJ and TEJ cultivated rice, we found 133 significant $F_{ST}$ outliers across the genome. A subset of 66 of these outliers was also detected in the BH/TEJ comparison. Interestingly, these critical weed-associated variants appeared to occur in genomic clusters within the genome: 29 (43.9%), 10 (15.2%) and 9 (13.6%) of the outliers occurred on chromosomes 7, 1 and 2, respectively. This finding of 'de-domestication blocks' within the genome aligns well with

prediction that most weediness adaptation occurs through changes in relatively few genomic regions[7,11].

Shared differentiated regions in BH and SH_TEJ weedy rice include *Rc* and *GD1* (regulating seed germination and seedling development[33]) on chromosome 7, as indicated in a previous study[11]. The two weedy rice strains also shared differentiated regions, including *KO1* and *KO2* (regulating GA biosynthesis) on chromosome 6, *PLT1* (regulating root development[31] and stress response[32]) on chromosome 4, and *ERF3* and *ASR1* (regulating drought tolerance) on chromosomes 1 and 7, respectively (Fig. 3a, b). These patterns further support weed adaptation through mechanisms involving seed dormancy and stress tolerance as indicated by the shared adaptation mechanisms in BH and SH_TRJ weedy rice.

For the distinct differentiated regions in BH and SH_TEJ weedy rice, SH_TEJ carries cultivated rice alleles (the same alleles as the Nipponbare reference) for genes associated with plant height, such as *FC116*, *KO1* and *KO2* (Fig. 4a, Supplementary Table 2; Supplementary Data 3). The region including *FC116* was not found to have significant outliers when comparing SH_TEJ and TEJ, indicating that these alleles inherited from the crop ancestor could have been fixed by genetic drift. In contrast, *KO1* and *KO2* were located in significantly differentiated regions on

chromosome 6 when comparing SH_TEJ and TEJ, suggesting that positive selection has favoured alleles inherited from the crop ancestor.

**Signatures of positive selection on the targets for parallel evolution**. To further substantiate signatures of positive selection, we performed selection scans based on extended haplotype homozygosity (EHH)[34]. A cross-population EHH (XP-EHH) test[35] supported the signatures of adaptive differentiation between weed strains and their inferred cultivated ancestors identified by $F_{ST}$ in BH and SH_TEJ weedy rice: most of the differentiation regions were also confirmed by the XP-EHH test (Supplementary Fig. 8a and b). For SH_TRJ weedy rice, the signatures of differentiation were not confirmed by XP-EHH tests in several regions, but some regions regulating seed dormancy (*LOL1* on chromosome 8) and drought tolerance (*CUL1-1* and *CUL1-3* on chromosome 1) were also considered significantly differentiated regions between SH_TRJ weeds and TRJ cultivars (Supplementary Fig. 8c).

The integrated haplotype score (iHS) test, which detects positive selection signatures within populations, was conducted to further identify regions displaying strong signatures of positive selection within weedy rice groups. Among the candidate genes located in shared differentiated regions between BH and SH_TRJ or BH and SH_TEJ by $F_{ST}$ values, *LOL1* on chromosome 8 and *KO1* and *KO2* on chromosome 6 were included in the regions where significant signatures were detected by the iHS tests (Fig. 4b). *LOL1*[28] and *KO1/KO2*[27–29] influences GA biosynthesis, thereby regulating seed dormancy. All of the weedy rice strains from BH, SH_TEJ and SH_TRJ had the same allele for *LOL1*, but SH_TEJ had different alleles for *KO1/KO2* from the other weeds (Fig. 4a). Genomic scans with iHS thus confirm conserved and distinct mechanisms of de-domestication among weed strains, and, interestingly, some regions regulating GA biosynthesis and seed dormancy are detected in the independent feralization events.

**Local ancestry inference**. Because the SH_TEJ weed strains are admixed derivatives of BH weedy rice and TEJ cultivars, we applied local ancestry analysis to identify specific regions of the genome that were contributed by BH or TEJ. This analysis indicated that the proportion of ancestry was variable across chromosomes. A significant proportion of the SH_TEJ weedy rice genomes originated from BH weedy rice on chromosomes 1, 2 and 7, and originated from cultivated rice on chromosomes 4, 5, 6 and 8 (Fig. 5). Additionally, the highly consistent patterns of local ancestry across individuals within the subgroups (SH1_TEJ or SH2_TEJ) suggests that within each subgroup, all of the individuals are descended from a single admixture event. Notably, both of the SH weedy rice subgroups shared local ancestry from BH weedy rice in some regions, such as chromosomes 1, 2 and 7; this suggests that the regions corresponding to the de-domestication blocks identified by $F_{ST}$ outlier analysis have been selectively retained in both SH_TEJ subgroups since the crop-weed hybridization events that gave rise to them.

## Discussion
In this study, we found evidence of conserved mechanisms of parallel weediness evolution using weedy rice strains that evolved independently from different cultivated rice subgroups. We also documented novel mechanisms of weediness evolution: weedy rice gained a wild-like trait (seed dormancy) from cultivated rice through hybridization. In general, weedy rice has evolved multiple times independently from domesticated ancestors, and most Old World strains have in situ origins within the regions where

they occur[7,8,10,11,36]. Japanese weedy rice evolved from East Asian *japonica* cultivated rice[10,11], and this study has revealed that BH and SH_TRJ weedy rice evolved from East Asian TEJ or TRJ cultivated rice, respectively. This finding provides a unique opportunity to reveal the diverse genetic mechanisms underlying weediness evolution from the *japonica* (TEJ and TRJ) subgroup.

Most of the genomic regions with signatures of positive selection are distinct between TEJ and TRJ weedy rice strains, and the distinct genetic mechanisms include the seed germination-related gene, *GD1*. This gene could be a target of parallel selection among different TEJ-derived weedy rice populations and may play a crucial role for the distinct germination behaviour[11], although our analysis reveals the gene is not under selection in TRJ-derived weedy rice. Here we found a few shared genomic regions with signatures of positive selection in both types of weedy rice, in addition to *Rc* which is well known as a shared target of selection in weed evolution[7,11] and which pleiotropically controls both red pericarp and seed dormancy[30]. One of the shared genes detected in this study is *LOL1*, which regulates GA synthesis and seed dormancy. Interestingly, weedy rice strains harbour the same allele of *LOL1* as modern cultivated rice varieties, but some landrace varieties have different alleles (Fig. 4a). Some Asian weedy rice strains appear to have very recent origins from elite cultivars[11], and our results suggest that feralization from modern varieties may occur more easily than that from landraces that do not have the *LOL1* allele.

Our analyses also revealed that hybridization plays important roles in the evolution of weedy rice. SH_TEJ weedy rice strains originated from hybridization between BH weedy and cultivated rice, with TEJ cultivars as the maternal ancestor (Fig. 2). Adaptive introgression of crop-like traits, such as plant height and herbicide resistance, is considered an important consequence of hybridization with cultivated rice, and wild-like traits, such as easy seed shattering and deep seed dormancy, are considered important traits for adaptive introgression from wild rice[37]. Some studies have revealed introgression of a crop-like trait (herbicide resistance[16–18]) from cultivated rice through hybridization; interestingly our study shows introgression of a wild-like trait—seed dormancy—from cultivated rice. SH_TEJ weedy rice strains have deeper seed dormancy and lower plant height than BH weedy rice strains (Supplementary Fig. 7), and we detected genetic contributions of introgressed alleles at the *KO1* and *KO2* loci from cultivated rice for weediness traits, including wild-like traits (deep seed dormancy) and crop-like traits (short plant height). *KO1* and *KO2*, regulating GA biosynthesis, pleiotropically control seed dormancy and plant height with their mutants showing deep seed dormancy and semidwarf plant height[27–29].

Some associations between seed dormancy and plant height have been reported; for example, semidwarf genotypes have deeper seed dormancy than tall genotypes, and *semidwarf 1* (*sd1*), associated with GA biosynthesis, regulates seed dormancy and semidwarf plant height[38,39]. Seed dormancy in cereal crops should be moderate (not too much, but not lacking altogether) because lack of seed dormancy causes preharvest sprouting which significantly reduces the yield and quality of cereal crops[40,41]. The introduction of semidwarf traits into cereal crops was one of the most important strategies deployed in modern breeding because it increases lodging resistance and a greater harvest index, allowing for the application of large amounts of nitrogen fertilizer[42]. So, artificial selection during rice breeding has kept some alleles related to deep seed dormancy and semidwarf plant height for preventing preharvest sprouting and lodging, suggesting that crop may become a source of deep seed dormant traits. SH_TEJ weedy rice strains also have more persistence in the soil seedbank than BH weedy rice strains[43], so hybridization

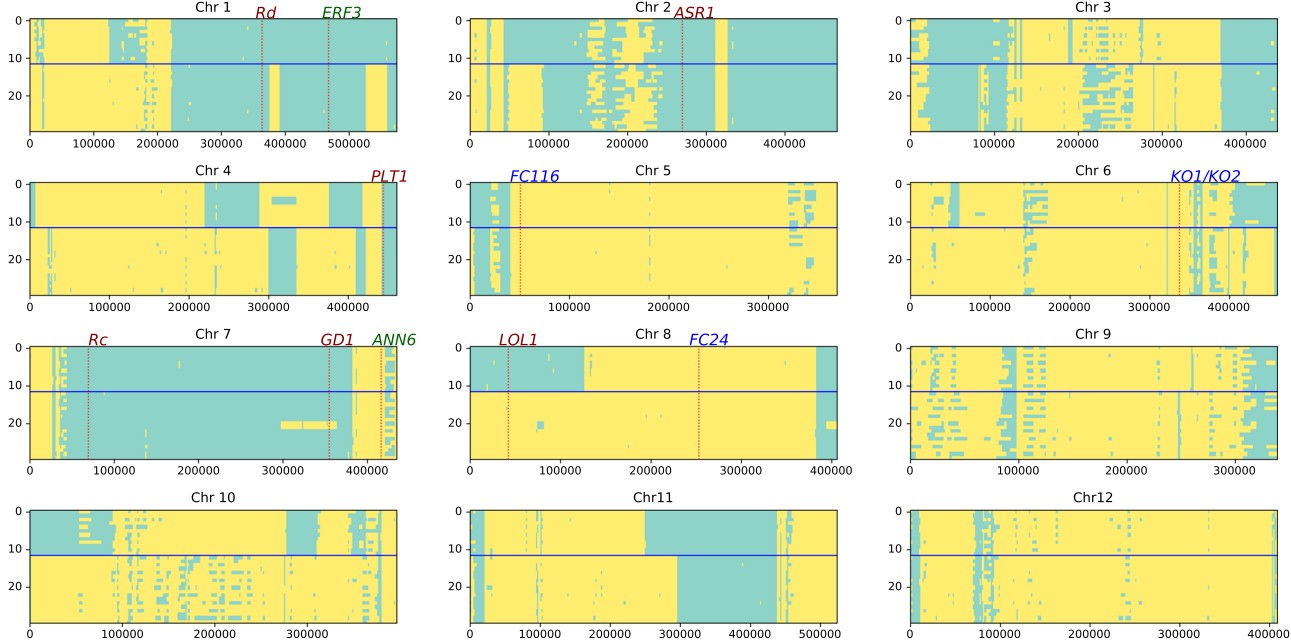

**Fig. 5 Local ancestry inference for 15 SH_TEJ weedy rice individuals.** Each panel corresponds to a single chromosome, with each individual represented by two rows. The top six individuals (12 rows) are SH1_TEJ individuals, and others are SH2_TEJ individuals. The x-axis indicates the number of SNP sites examined (not their physical location). The functionally known genes indicated in Fig. 3b and genes regulating plant height (FC116 and FC24) are shown. Green-cyan: BH weedy rice ancestry, yellow: cultivated rice ancestry.

with cultivated rice may have enhanced the competitive ability of weedy rice.

Our whole-genome sequence analyses reveal that repeated evolution of weedy rice occurred through different genetic mechanisms within Japan, indicating that diverse weedy rice strains can easily evolve even within geographically restricted areas. Moreover, some weedy rice strains gained not only crop-like adaptive traits but also weedy-like adaptive traits through hybridization. Adaptive introgression from crops to weeds or wild-relatives has been detected for some crop-like traits, such as herbicide resistance[16–18] and a flowering time gene which makes weeds complete their life cycle during the cropping season[44]. Here we show that adaptive introgression from crop could be an important source of novel genetic variation of weedy-like adaptive traits. Intra- and inter-population diversity is important for adaptation in agricultural weeds[45], so multiple de-domestication routes, including hybridization with cultivated rice, with a few genetic constraints make weedy rice especially difficult to control.

## Methods

**Sampling and phenotypic classification.** A total of 50 plant samples representing 48 populations of Japanese weedy rice (each corresponding to an individual paddy rice field) distributed across 11 prefectures were collected from 2014 to 2017 (Supplementary Data 1). All weedy rice strains used in this study had a red pericarp colour and easily shattered seeds. Accessions were classified by hull colour (as either BH or SH) and apiculus colour (either red or colourless) (Supplementary Data 1). In addition to the weedy rice samples, five weedy rice strains collected in Japan in the 1970s and 33 Japanese landraces were analysed (Supplementary Data 1). Seeds for all strains of the 1970s weedy rice and landraces were obtained from the Genetic Resources Center or Central Region of Agricultural Research Center, National Agricultural and Food Research Organization (NARO), Japan or Institute of Genetic Resources, Faculty of Agriculture, Kyushu University, Japan.

For phenotypic classification, we evaluated culm length and seed dormancy. Culm length was evaluated during heading for 6 individuals per strain that were grown in an experimental field at NARO (Tsukuba, Japan) in 2019. Eleven BH weedy rice strains, 5 SH_TEJ weedy rice strains, and one modern_TEJ cultivated rice strain (Koshihikari) were used to compare culm length among subgroups. For the evaluation of seed dormancy, germination assays were conducted on seeds after-ripened for 1, 5 and 10 weeks after collection from farmers' fields in 2016. Eight BH and 5 SH_TEJ weedy rice strains were used, and seeds were incubated under two conditions (15 °C dark or 30 °C dark) with four replicates. Each replicate

comprised 20 seeds sown in 55 mm Petri plates with 5 ml of distilled water. Seeds were censused regularly, germinated seeds were removed, and the number of seeds that germinated was counted until 14 days. The final germination rate of viable seeds that germinated was used for analysis, with each plate being the unit of analysis. Germination rates were estimated by a generalized liner mixed model with a binomial link, with strain included as a random effect using R ver. 3.63[46].

**Sequencing and genotype calling.** We sequenced the genomes of 50 Japanese weedy rice strains collected from paddy rice fields, five Japanese weedy rice strains collected in the 1970s, and 33 landrace strains, at 22.6× average genome coverage (Supplementary Data 1). DNA libraries were constructed by BGI (Hong Kong, China) and sequenced with the Illumina HiSeq X Ten platform. The short-read sequence data have been deposited into the DRA/DDBJ repository under Bio-Project accession numbers PRJDB8989 (DRR209178-DRR209229) and PRJDB10870 (DRR255840-DRR258879). The 50 weedy rice strains deposited as PRJDB8989 were previously analysed in a comparative study of weedy rice strains worldwide[11]. Other genomic data from 86 cultivated and weedy rice strains (Supplementary Data 1) were obtained from previous studies to further represent Japanese cultivated rice and non-Japanese weedy rice[7,8,47–52]. The downloaded Japanese landrace strains were from the TEJ, TRJ or IND group (hereafter, landrace_TEJ, landrace_TRJ or landrace_IND, respectively), and the downloaded Japanese modern strains were from the TEJ or IND/admix group (hereafter, modern_TEJ or modern_IND/admix, respectively). The downloaded non-Japanese weedy rice strains were from the Chinese TEJ or IND group and US IND or AUS groups (hereafter, China_TEJ, China_IND, US_IND or US_AUS, respectively). Three Chinese landraces were also downloaded to represent the genetic diversity of Chinese crops in a previous study[8]. We selected 86 published genomes on the basis of high genome coverage; 78% of the selected strains had >15× genome coverage (the minimum genome coverage was 11.8 ×).

The raw paired-end reads were first trimmed with Trimmomatic version 0.38[53] with parameters 'ILLUMINACLIP:2:30:10 LEADING:20 TRAILING:20 SLIDINGWINDOW:10:20 MINLEN:30'. Then, the trimmed reads were aligned to Os-Nipponbare-Reference-IRGSP-1.0[54] pseudomolecules using BWA-MEM (release 0.7.17[55]). The mapped reads were then sorted and duplicates were removed by Picard tools (release 2.18.17[56]). The variants were called for each accession by the GATK (release 4.0.11.0[57]) HaplotypeCaller followed by genotyping with the GATK GenotypeGVCFs. To reduce false positives, the SNP calls were filtered according to the following thresholds 'QD < 5.0, FS > 50.0, SOR > 3.0, MQ < 50.0, MQRankSum < −2.5, ReadPosRankSum < −1.0, ReadPosRankSum > 3.5'. Imputation of all reported SNPs in the 174 strains from variant call format (VCF) was performed by Beagle v5.0[58] with default parameters using the genotype likelihoods. Then, we converted the imputed VCF data to FASTA format using in-house Perl scripts. A set of 1,972,399 SNPs that were homozygous and without missing data across the 174 rice samples were used to construct a neighbour-

joining tree with MEGA7[59]. The phylogenetic trees were visualized using ggtree v1.1.6[60] in R 3.6.3[46].

To elucidate maternal ancestry based on chloroplast genome sequences, we extracted variants (SNPs and insertions/deletions(INDELs)) in the chloroplast genome sequences (VCF file before imputation) for 147 TEJ and TRJ cultivated and weedy rice strains. Seven variants were detected in the chloroplast genome sequences of 130 of the 147 strains; the remaining 17 strains were not included because of missing data at the variant sites.

**Analysis of genetic diversity and population structure**. PCA was conducted to assess the genetic relationships within and among the study populations using PCAngsd ver 0.97[61]. We also estimated ancestry proportions for individuals using NGSadmix[62] using the genotype likelihoods, which implements a clustering method similar to that in the population program ADMIXTURE[63] while incorporating uncertainty inherent in the genotype calls inherent in next-generation sequencing data. NGSadmix runs were performed for K values ranging from 2 to 10 with default parameters in NGSadmix. The optimal K was estimated using the $\Delta K$ statistic[23] as the selection criterion. To further assess the relationship of weedy rice to cultivated rice strains, Kinship analysis was conducted using NGSrelate[64] using the same samples as used in PCAngsd and NGSadmix analysis, plus 11 cultivated rice strains (Supplementary Data 1) that have been shown to have high kinship with Japanese weedy rice strains in the previous study[11]. Genomic data of the 11 cultivated rice strains were obtained from previous studies[65,66]. PCAngsd, NGSadmix and NGSrelate use genotype likelihoods, thereby accounting for the uncertainty in the called genotypes that is inherently present in low-depth sequencing data[61]. The genotype likelihoods were estimated by ANGSD[67] with BAM files after removed duplication by Picard tools as above. All of the analyses were conducted with default parameters as described in manuals[68–70].

**Identification of genomic regions under selection**. The genome was scanned in 100-kb windows with a step size of 10 kb, and the population differentiation index $F_{ST}$ was estimated for each window by VCFtools. In the genomic differentiation analyses based on the population differentiation index ($F_{ST}$), we compared weedy rice of one group against the totality of its potential ancestors (e.g., BH weedy rice vs. TEJ modern and landrace varieties). Z-transformation was applied to locate divergent regions between weedy rice and cultivated rice from the extreme tails by applying a threshold of five absolute median deviations (at least five deviations from the median).

The EHH test was also conducted to identify genomic regions under selection and was implemented in the R package rehh 3.1.2[71] in R 3.6.3[46]. We selected SNPs within the TEJ (BH, SH_TEJ weedy rice and TEJ modern/landrace varieties) and TRJ (SH_TRJ weedy rice and TRJ landrace variety) groups, and a total of 1,243,658 and 1,594,881 SNPs were used for the EHH test, respectively. We performed the EHH test based on XP-EHH[35] to detect genomic differentiation between populations and iHS[72] to detect selective sweeps within populations using the parameters as described in a manual[73]. The XP-EHH and iHS values were applied to locate regions under positive selection by applying a threshold of $p < 0.0001$ ($-\log10(p$-value) > 4.0).

**Demographic history of weedy rice**. To investigate how demographic processes have influenced the evolution of TEJ-derived weedy rice from Japan, we used an approximate Bayesian computation (ABC) approach implemented in the program DIYABC v. 2.1.0[74]. We compared four demographic scenarios: (1) BH weedy rice and Japanese cultivated rice evolved from a shared ancestor, after which SH weedy rice evolved from a hybridization event between BH weedy and cultivated rice; (2) Japanese cultivated rice and the shared weedy rice ancestor of BH and SH diverged at first, after which BH and SH weedy rice diverged from the shared ancestor; (3) BH weedy rice diverged from a shared ancestor earlier than SH weedy rice; and (4) SH weedy rice diverged from a shared ancestor earlier than BH weedy rice. All analyses were based on a subset of 4,289 SNPs extracted from 212,319 homozygous SNPs among Japanese TEJ cultivated rice and TEJ-derived weedy rice using VCFtools with 10,000 thinned methods (extracting one SNP per 10,000 bp). Demographic scenario selection and parameter estimates were based on a total of four million simulations (one million per scenario), as suggested by the DIYABC documentation. Posterior probabilities of the four scenarios were calculated by logistic regression considering 40,000 datasets that were closest to the observed values.

Then, we rebuilt three demographic scenario models involving BH and SH weedy rice and modern and landrace cultivars to deduce the hybridization history of SH weedy rice: (1) SH weedy rice evolved from a hybridization event between BH weedy rice and cultivated rice before the divergence of modern and landrace strains; (2) it evolved from a hybridization event between BH weedy and modern cultivated rice; and (3) it evolved from a hybridization event between BH weedy and landrace cultivated rice. To further assess the goodness-of-fit of each demographic scenario model, we performed model analysis with default parameters in DIYABC v. 2.1.0. If the model fit the data well, the observed data were plotted within the scope of simulated datasets of posteriors in the PCA plots.

The SH_TRJ evolution was analysed based on a subset of 2591 SNPs, and demographic scenario selection was based on a total of six million simulations. Five

SH_TRJ weedy rice strains and 10 landrace_TRJ strains having haplotype 2 of chloroplast genome (Supplementary Data 1) were used for the analysis.

**Local ancestry inference**. Phased haplotypes were imputed from the merged dataset separately for the 15 SH weedy rice strains and their source populations (BH weedy rice and TEJ modern cultivated rice) using Beagle v5.0 with default parameters. The sample size of putative source populations affects subsequent local inference; therefore, we randomly selected 10 accessions of BH weedy rice to match the 10 TEJ modern cultivated rice samples. Using the BH weedy rice and TEJ modern varieties as putative source populations, the local ancestry of the SH weedy rice haplotypes was inferred with Loter[75]. Loter does not require specifications of uncertain biological parameters (genetic maps, recombination and mutation rates, average ancestry coefficients, or number of generations since admixture) and has been shown to outperform other tools in local ancestry inference of ancient admixture events[75].

**Statistics and reproducibility**. This study is based on the short-read whole-genome sequencing of 174 weedy and cultivated rice strains, and all population genomic analyses were performed using publicly available software described in Methods. All statistical analyses were preformed using R ver. 3.63. Reproducibility including parameters for the population genomic analyses, sample sizes and number of replicates are stated in Methods and Figures.

**Reporting summary**. Further information on research design is available in the Nature Research Reporting Summary linked to this article.

## Data availability
Short-read sequence data of 88 samples generated by this study have been deposited into DDBJ under the bioproject accession number PRJDB8989 (DRR209178-DRR209229) and PRJDB10870 (DRR255840-DRR258879). All other sequence data needed to evaluate the conclusion are indicated in Supplementary Data 1.

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

## Acknowledgements

The authors thank the staff of the Yamagata Prefectural Government, Fukushima Prefectural Government, Tochigi Prefectural Government, Ibaraki Agricultural Institute, Chiba Prefectural Government, Yamanashi Prefecture Agricultural Research Center, Nagano Prefectural Government, Mie Agricultural Research Institute, Shiga Prefectural Government, Yamaguchi Prefectural Government, and Miyazaki Prefectural Government for providing assistance in the field collection of weedy rice. We also thank Institute of Genetic Resources, Faculty of Agriculture, Kyushu University for providing some landrace varieties. This research was supported by the Advanced Analysis Center Research Supporting Program of National Agriculture and Food Research Organization (NARO) and the Advanced Genomics Breeding Section of Institute of Crop Science, NARO (NICS). We also thank the Advanced Analysis Center of NARO (NAAC) and AFFRIT of Ministry of Agriculture, Forestry and Fisheries(MAFF), Japan for use of the high performance cluster computing system, respectively. The authors thank Maiko Akasaka (NARO), Atsushi J. Nagano and Ayumi Deguchi (Ryukoku University) for providing informative insights and knowledge, and thank Gabriela A. Auge (IIBBA-CONICET) for helpful comments on earlier versions of this manuscript. This work was supported by grants from the Project of the Bio-oriented Technology Research Advancement Institution, NARO (the special scheme project on advanced research and development for next-generation technology), and from the commissioned project "Development of labour-saving management of serious weeds to expand cultivation of direct-seeded rice", MAFF, Japan.

## Author contributions

T.I., H.K. and A.K. designed and managed the project. T.I., K.E. and C.M. collected the samples and carried out the experiments. T.I. and Y.K. performed the bioinformatic analyses. K.M.O. gave insightful suggestions and comments on the bioinformatic analyses. T.I. and K.M.O. wrote most of the manuscript. All authors read and approved the final manuscript.

## Competing interests

The authors declare no competing interests.
