## [Peer Review File · Communications Biology]

Editorial Note: This manuscript has been previously reviewed at another journal that is not operating a transparent peer review scheme. This document only contains reviewer comments and rebuttal letters for versions considered at Communications Biology.

Reviewers' Comments:

Reviewer #1:

Remarks to the Author:

This paper looks at the origin and evolution of Japanese weedy rice. The authors find that there is parallel evolution in rice feralization, and document the temperate and tropical japonica origins of weedy rice in Japan. The authors also document the role of hybridization in weedy rice origins.

In general this is a fairly systematic and comprehensive analysis of the evolution of this group of weedy rice in Japan. The authors are also able to document diverse origins as well as selection on several loci; the selection analysis is fairly noteworthy as it provides a good indication of the genetic basis for selection between cultivated and weedy rice. The techniques carried out are fairly well-known and I do not see major issues in the analysis and conclusions.

Reviewer #2:

Remarks to the Author:

In this manuscript, authors reported that repeated evolution of weedy rice through different genetic mechanisms in Japan, and that some Japanese weedy rice have evolved through adaptive introgression from cultivated rice for both crop-like traits and weedy-like traits. This is an important finding as it shows that when a standing weedy rice hybridizes with cultivated rice, the progeny may be weighted toward the weedy direction and becomes even weedier. The data are sound and well analyzed with appropriate statistical supports. The conclusions are strongly supported by data. The methods section are attentive to details enable reproducible analyses. The manuscript is well-written with clear logic and easy to follow.

This reviewer evaluated an earlier version of the manuscript submitted to another journal. Compared with the earlier version, the manuscript is substantially improved. Especially, my comments about the importance to clarify of novelty of this study by comparing with existing studies have been satisfactorily done. As such, I do not have additional concerns. This study has clear novelty and extends our understanding of the complex genetic mechanisms underlying rice feralization, an emergent issue in rice production. It will be interested by the weedy rice community and beyond. I fully endorse publication of this excellent study by Communications Biology.